# Quantifying and Defending against the Privacy Risk in Logit-based Federated Learning

## Abstract

Federated learning (FL) aims to protect data privacy by collaboratively learning a model without sharing private data among clients. Novel logit-based FL methods share model outputs (i.e., logits) on public data instead of model weights or gradients during training to enable model heterogeneity, reduce communication overhead and preserve clients' privacy. However, the privacy risk of these logit-based methods is largely overlooked. To the best of our knowledge, this research is the first theoretical and empirical analysis of a hidden privacy risk in logit-based FL methods – the risk that the semi-honest server (adversary) may learn clients' private models from logits. To quantify the impacts of the privacy risk, we develop an effective attack named Adaptive Model Stealing Attack (AdaMSA) by leveraging historical logits during training. Additionally, we provide a theoretical analysis on the bound of this privacy risk. We then propose a simple but effective defense strategy that perturbs the transmitted logits in the direction that minimizes the privacy risk while maximally preserving the training performance. The experimental results validate our analysis and demonstrate the effectiveness of the proposed attack and defense strategy.

## 1 Introduction

In recent years data privacy regulations such as General Data Protection Regulation (GDPR) have largely restricted the collection of annotated data on individuals for centralized training. Federated Learning (FL) (McMahan et al., 2017) provides a promising approach that allows different clients to collaboratively train their models by sharing local model parameters or gradients without exchanging their respective raw data. However, recent studies (Zhu et al., 2019; Geiping et al., 2020) reveal that private training data can be derived from the shared gradients or parameters, which poses serious privacy leakage issue.

Another line of FL studies (Chang et al., 2019; Gong et al., 2021; 2022; Jeong et al., 2018; Li & Wang, 2019) adopt knowledge distillation (Hinton et al., 2015) to exchange model outputs (i.e., logits) instead of model weights or gradients during training to reduce communication overhead and enable model heterogeneity. To preserve clients' privacy, these logit-based FL methods distill on public data to transfer knowledge and model parameters are stored locally during training, as depicted in Figure 1. Moreover, such public data can be unlabeled and insensitive that is sampled from other domains (Gong et al., 2022).

A natural question arises: Is the logit-sharing scheme safe to protect the privacy of each participant? Unfortunately, we find that the transmitted logits can still pose the privacy leakage risk, e.g., the adversary may learn clients' private models by leveraging its predictions on public data among the training iterations. Such leakage poses intellectual property issues and can serve as a stepping stone for further attacks, such as membership inference attacks (Nasr et al., 2019) and data reconstruction attacks (Geiping et al., 2020; Zhu et al., 2019).

To the best of our knowledge, we are the first to provide a theoretical and empirical analysis of a hidden privacy risk in logit-based FL that the semi-honest server intends to infer clients' private models without knowing local model architecture or data distribution. To quantify the impacts of the privacy risk, we develop an effective attack named Adaptive Model Stealing Attack (AdaMSA), which adaptively steals the private model by approximating its intermediate training states in previous iterations. Specifically, in each iteration, the semi-honest server forces the attacking model to

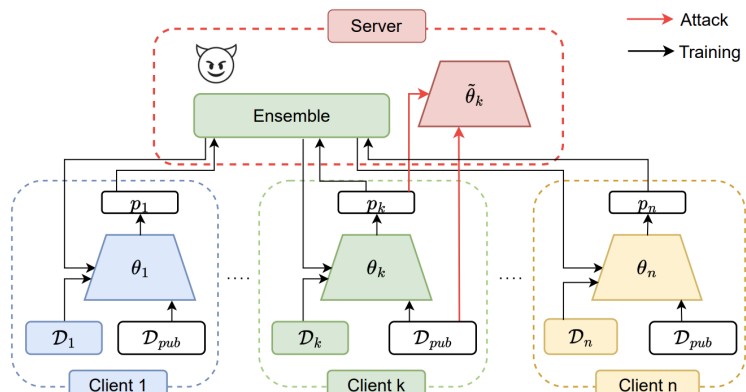

Figure 1: Illustration of logit-based FL and our attack setting. The server is semi-honest and aims to infer client $k's$ private model via its predictions $p_k$ on an unlabeled public dataset $\mathcal{D}_{pub}$ during training.

approximate the current state of the victim model by minimizing the distance between the output of the attacking model and a target logit (i.e. attacking logit). Inspired by ensemble learning (Mienye & Sun, 2022), we propose to combine the observed historical logits of the victim model via an importance weight to obtain a more informative attacking logit. Additionally, we provide a theoretical analysis on the bound of this privacy risk in logit-based FL.

Moreover, we propose a simple but effective perturbation-based defense strategy to prevent this privacy leakage in logit-based FL. The key idea of our strategy is to perturb the logit in the direction that maximally thwarts the adversary while minimally reducing the model performance loss. As a result, our defense achieves a better trade-off compared to prior art.

We empirically evaluate our proposed attack and defense in three experimental settings, Close-world, Open-world-CF and Open-world-TI (see Section 5.1 for details). The experimental results show that AdaMSA is effective and our defense can offer a better utility and privacy trade-off than the state-of-the-art baselines. We believe that our research can shed light on the hidden privacy risk of logit-based FL and pave the way toward privacy-preserving FL methods.

Our key contributions are summarized as follows:

- To the best of our knowledge, we provide the first theoretical and empirical analysis of a hidden privacy risk in logit-based FL that the semi-honest server can infer clients' private models according to logits.

- To quantify the privacy risk, we develop an effective model stealing attack named AdaMSA, which steals private models by leveraging historical logits during training. Moreover, we provide a theoretical bound for the privacy risk.

- To prevent the privacy risk, we develop a simple but effective defense by perturbing the transmitted logits in the direction that minimizes the privacy risk while maximally preserving the training performance.

- We empirically evaluate our designed attack and defense in three experimental settings, Close-world, Open-world-CF and Open-world-TI. The results validate our analysis, show that AdaMSA can achieve up to 3.69% improvement and our defense can achieve a better utility and privacy trade-off compared to the state-of-the-arts.

## 2 RELATED WORK

### 2.1 PRIVACY RISK IN FEDERATED LEARNING

Federated learning (Kairouz et al., 2019) allows multiple clients to collaboratively train a global model while keeping training data locally. Typical FL algorithms (McMahan et al., 2017; Karimireddy et al., 2020) are parameter-based FL that shares local model parameters or gradients and

aggregate local models in the server. Logit-based FL (Chang et al., 2019; Gong et al., 2021; 2022; Jeong et al., 2018; Li & Wang, 2019) adopt knowledge distillation (Hinton et al., 2015) to transmit model outputs (i.e., logits) instead of model weights or gradients during training to reduce communication overhead, enable model to be heterogeneous and preserve clients' privacy.

Previous studies have thoroughly analyzed the privacy risks of sharing model parameters or gradients in FL, including class representatives leakage (Wang et al., 2019), membership leakage (Nasr et al., 2019), property leakage (Melis et al., 2019) and training input leakage (Geiping et al., 2020; Zhu et al., 2019). However, these efforts are all white-box attacks (detailed comparisons are summarized in Appendix A). That is, they have strong assumptions that the adversary knows the local model architecture and detailed training information such as gradients. In this work, we focus on logit-based FL that model parameters or gradients are stored in clients' local machines. To our knowledge, this is the first study analyzing the privacy risk in logit-based FL.

## 2.2 Model Stealing Attack

Model stealing attacks (Orekondy et al., 2019; Papernot et al., 2017; Tramèr et al., 2016) have demonstrated the ability to steal a deployed machine learning model in a black-box manner through limited query access and carefully calibrated proxy dataset. These attacks happen in the inference stage and aim to reduce the number of queries or eliminate the need of proxy dataset. However, in logit-based FL, the attack happens in the training stage, where the adversary can neither arbitrarily select the query dataset nor access to the private models or private dataset distribution. Instead, the adversary only observes the intermediate information (i.e., transmitted logits) from the victim during training. Based on this observation, we propose AdaMSA that leverages historical training information to obtain more informative attacking logits and therefore improve the attack performance.

## 2.3 Privacy Protection Strategy in Logit-based FL

Researchers have proposed some strategies (Li & Wang, 2019; Gong et al., 2022; Sattler et al., 2021) to prevent the potential privacy leakage in logit-based FL. Specifically, Li et al. (Li & Wang, 2019) proposed to distill on a public dataset instead of private data to transfer predicted vectors. Gong et al. (Gong et al., 2022) further relaxed the public data to be unlabeled and insensitive data sampled from other domains to preserve data privacy. Moreover, Sattler et al. (Sattler et al., 2021) and Gong et al. (Gong et al., 2022) adopted differential privacy (DP) to protect the transmitted logits. However, these paper fails to quantify the privacy risk inside logit-based FL and their defense strategies incur a significant loss in accuracy. In contrast, we first identify and quantify the privacy risk. Then we design a simple but effective perturbation strategy against our revealed privacy risk, which perturbs the logit in the direction that maximally misleads the adversary while minimally persevering training performance. Therefore, it can achieve a better utility and privacy trade-off.

## 3 Quantifying the Privacy Risk

In this section, we first give the problem setup and threat model. Then we propose an attack to quantify the privacy risk in logit-based FL and elaborate our proposed attack in details. Lastly, we give a theoretical analysis on the bound of this privacy risk.

### 3.1 Problem Setup and Threat Model

**Problem Setup:** As shown in Figure 1, there are $n$ clients and a central server. Each client has a private labeled dataset $\{\mathcal{D}_i\}_{i=1}^n$ and some unlabeled public data $\mathcal{D}_{pub}$. The server coordinates the training process, aggregating the client's submitted locally predicted logits $\{p_i\}_{i=1}^n$ on public data to obtain an ensemble logit $p_e$ and distribute it back to clients. Then clients train their local models $\{\theta_i\}_{i=1}^n$ under supervision of labels on $\{\mathcal{D}_i\}_{i=1}^n$ and the ensemble logit on $\mathcal{D}_{pub}$.

**Threat Model:** We assume that the server (i.e., adversary) is semi-honest, i.e. it completes the learning task as required but is curious about the clients' local model, and all clients are honest. The adversary does not know the private data distribution and the victim model, including its parameters, hyperparameters or architecture. Moreover, the adversary does not have the right to select public data. The adversary only knows that 1) the unlabeled public dataset $\mathcal{D}_{pub}$; 2) the transmitted victim's

logits $\{p_t\}_{t=1}^T$ on the public data during training. The adversary's goal is to steal the functionality of the victim's model $f(x, \theta)$ by training an attacking model $\tilde{f}(x, \tilde{\theta})$ that achieves high classification accuracy on the victim $k$'s private dataset $\mathcal{D}_k$:

$$\max_{\tilde{\theta}} \mathbb{E}_{x \sim \mathcal{D}_k} Acc(\tilde{f}(x, \tilde{\theta})).$$

In the following discussion, we assume that the adversary is interested in client $k$'s model and denote it as $\theta$ for simplicity.

| Threat Model | Adversary | Attack Target | Adversary's Knowledge |
|---|---|---|---|
| Semi-honest | Server | Private model $\theta$ | Logits of $\theta$ on $\mathcal{D}_{pub}$ during training and $\mathcal{D}_{pub}$ |

Table 1: Threat model.

## 3.2 ADAPTIVE MODEL STEALING ATTACK

To quantify the privacy risk in logit-based FL, we develop an Adaptive Model Stealing Attack named AdaMSA, which adaptively steals the private model by approximating its intermediate training states in previous iterations. Specifically, in each iteration, the server forces the attacking model to approximate the victim model by imitating a target logit (i.e. attacking logit) on public data. Since a more informative attacking logit will provide better supervision for the attacking model, the key issue here is how to design the attacking logit.

Given the historical logits during training, we design the attacking logit based on two considerations: 1) historical predictions may contain valuable information and can provide different views of data to improve generalization ability of student model (Allen-Zhu & Li, 2020); 2) predictions in the early rounds may not be well-trained to provide informative supervision. Accordingly, we define the attacking logit $\hat{p}_T$ as

$$\hat{p}_T = \sum_{t=T-T_0}^{T} w_t \cdot p_t,$$

where $w_t$ is an importance weight that represents how much attention we give to the past predictions and $T_0$ is a threshold that controls how far the past predictions we need to consider.

Considering that the victim model continuously evolves during the training process, we should give more attention to the prediction closer to the current iteration. Consequently, we set $w_t$ increase with iteration number $t$:

$$w_t = w_0 \cdot \frac{t}{T},$$

where $w_0$ is a normalization parameter.

To train the attacking model $\tilde{\theta}$, we minimize an empirical loss $L_{emp}$ computed between the attacking logit $\hat{p}_T$ and its prediction $\tilde{p}_T$ on $\mathcal{D}_{pub}$, which can be formulated as:

$$\min_{\tilde{\theta}} \mathbb{E}_{x \sim \mathcal{D}_{pub}} [L_{emp}(\hat{p}_T, \tilde{p}_T)].$$

We provide the algorithm of our proposed attack in one training iteration in Appendix B.1. By repeating this process, the semi-honest server is able to steal any interested intermediate private model during training as well as the final well-trained private model of the victim. The obtained attacking model with higher accuracy indicates that more privacy of the victim has been leaked.

## 3.3 ANALYSIS OF THE PRIVACY RISK

Logit-based FL methods transfer knowledge through the transmitted logits on public data during training. As shown in Figure 2 (b-c), we identify that the correlation (i.e. distance) between the private dataset and the public dataset plays a crucial role in determining the performance of logit-based FL methods. To better understand the inherent cause of privacy risk in this logit-sharing scheme, we start with quantifying the distance between the private dataset and the public dataset.

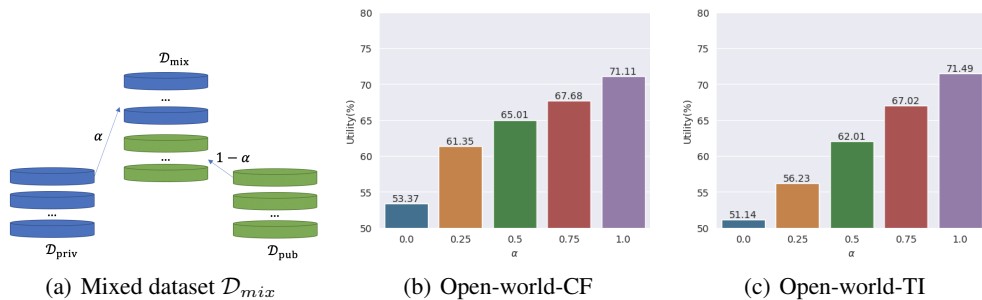

Figure 2: (a) Illustration of the mixed dataset $\mathcal{D}_{mix}$. (b-c) The impact of the distance between the private dataset and the public dataset on the utility of logit-based FL in Open-world-CF and Open-world-TI settings (See Section 5.1 for details). The distance is controlled by the weighting parameter $\alpha$.

Consider a simple case that we have n private datasets sampled from same distribution $\mathcal{D}_{priv}$ and an unlabeled public dataset from another domain sampled from independent distribution $\mathcal{D}_{pub}$. We construct several mixed datasets $\mathcal{D}_{mix}$ as the public dataset with the varying distance between private and mixed datasets controlled by a weighting parameter $\alpha$. As shown in Figure 2 (a), the mixed dataset is constructed through $\mathcal{D}_{mix} = (S_1, S_2)$, where $S_1$ consists of $\alpha|\mathcal{D}_{mix}|$ instances sampled independently from $\mathcal{D}_{priv}$ and $S_2$ consists of $(1-\alpha)|\mathcal{D}_{mix}|$ instances sampled independently from $\mathcal{D}_{pub}$. Through varying $\alpha$, we can thereby control the distance between private and mixed datasets, e.g. as $\alpha$ tends to 1, the mixed dataset approaches to the private dataset and vice versa.

As illustrated previously in Section 3.2, the privacy risk is measured by the performance of AdaMSA on the victim's private test dataset. In consequence, we can derive the bound of the privacy risk via the performance bound of AdaMSA on the victim's private test dataset based on the prior art in domain adaptation (Blitzer et al., 2007).

Denote the empirical risk of model $\theta$ on the mixed dataset as

$$\epsilon_{\mathcal{D}_{mix}}(\theta, f_p) = \mathbb{E}_{x \sim \mathcal{D}_{mix}}[|\theta(x) - f_p(x)|], \tag{1}$$

which measures the probability according the distribution $\mathcal{D}$ that $\theta$ disagrees with the ground truth label $f_p$. For simplicity, we abbreviate $\mathcal{D}_{mix}(\theta, f_p)$ as $\mathcal{D}_{mix}(\theta)$. Similarly, $\epsilon_{\mathcal{D}_{priv}}(\theta)$ and $\epsilon_{\mathcal{D}_{pub}}(\theta)$ denote the empirical risk of $\theta$ with respect to $\mathcal{D}_{priv}$ and $\mathcal{D}_{pub}$.

**Theorem 1.** Given a mixed dataset $\mathcal{D}_{mix} = (S_1, S_2)$, where $S_1$ consists of $\alpha|\mathcal{D}_{mix}|$ instances sampled independently from $\mathcal{D}_{priv}$, $S_2$ consists of $(1-\alpha)|\mathcal{D}_{mix}|$ instances sampled independently from $\mathcal{D}_{pub}$, its empirical risk can be written as

$$\epsilon_{\mathcal{D}_{mix}}(\theta) = \alpha\epsilon_{\mathcal{D}_{priv}}(\theta) + (1-\alpha)\epsilon_{\mathcal{D}_{pub}}(\theta).$$

The proof of Theorem 1 is given in Appendix D.1. Then we give some definitions and derive the bound of the difference between the empirical risk of $\theta$ on the mixed dataset $\mathcal{D}_{mix}$ and the private dataset $\mathcal{D}_{priv}$ for our analysis.

**Definition 1.** Given a domain $\mathcal{X}$ with $\mathcal{D}$ and $\mathcal{D}'$ probability distributions over $\mathcal{X}$, let $\mathcal{H}$ be a hypothesis class on $\mathcal{X}$ and $\mathcal{A}_{\mathcal{H}}$ be the set of subsets of $\mathcal{X}$ that supports the hypothesis in $\mathcal{H}$. The H-divergence between $\mathcal{D}$ and $\mathcal{D}'$ is defined as: $d_H(\mathcal{D}, \mathcal{D}') = 2\sup_{A \in A_H}|Pr_{\mathcal{D}}(A) - Pr_{\mathcal{D}'}(A)|$.

**Definition 2.** For a hypothesis space $\mathcal{H}$, the symmetric difference hypothesis space $\mathcal{H}\Delta\mathcal{H}$ is defined as $\mathcal{H}\Delta\mathcal{H} = \{h(x) \bigoplus h'(x)|h, h' \in \mathcal{H}$, where $\bigoplus$ represents the XOR operation.

**Theorem 2** ((Blitzer et al., 2007)). Let $h$ be a hypothesis in class $H$. Then we have

$$|\epsilon_{\mathcal{D}_{mix}}(h) - \epsilon_{\mathcal{D}_{priv}}(h)| \leq (1-\alpha)(\frac{1}{2}d_{\mathcal{H}\Delta\mathcal{H}}(\mathcal{D}_{priv}, \mathcal{D}_{pub}) + \lambda),$$

where $\lambda = \epsilon_{\mathcal{D}_{priv}}(h^*) + \epsilon_{\mathcal{D}_{pub}}(h^*)$ and $h^*$ is the ideal joint hypothesis minimizing the combined empirical risk: $h^* = argmin_{h \in H}\epsilon_{\mathcal{D}_{priv}}(h) + \epsilon_{\mathcal{D}_{pub}}(h)$.

The proof of Theorem 2 is given in Appendix D.2. In our case, the attacking model is trained on mixed dataset and test on victim's private dataset. According to Theorem 2, we obtain that the

bound of the privacy risk, measured by the performance of the attacking model $\tilde{\theta}$, is bounded by $(1-\alpha)(\frac{1}{2}d_{\mathcal{H}\Delta\mathcal{H}}(\mathcal{D}_{priv}, \mathcal{D}_{pub}) + \lambda)$. When fixing $\mathcal{D}_{priv}$ and $\mathcal{D}_{pub}$, this bound is only related to weighting parameter $\alpha$. This bound drops to 0 as $\alpha$ increases to 1. This indicates that, when $\alpha$ increases, i.e. the mixed public data gets closer to the private data, the attacking model performs better on the private test dataset and more private information of the model, which is trained on its logits and the public dataset, is leaked.

Here we briefly discuss the inherent cause of this privacy risk in logit-based FL. From Figure 2 (b-c), it is observed that local model training is benefited from the knowledge contained in the ensemble logit which is obtained through the ensemble of local predicted logits on the public data. Although a more informative local logit results in a more informative ensemble logit, it also exposes more privacy to the adversary as illustrated in Theorem 2. Our observation is also consistent with our empirical result as shown in Figure 3 in Section 5.2.

## 4 DEFENSE DESIGN

Our observation in Section 3 shows that the privacy risk in logit-based FL mainly comes from the logit. In this section, we propose a defense strategy that perturbs the transmitted logits of local models to defend against this privacy risk.

**Defense Objective** The defender (i.e. the clients) has two objectives. First, the defender aims to prevent an adversary from being able to replicate the functionality of its private model:

$$\min_{\tilde{\theta}}\mathbb{E}_{x\sim\mathcal{D}_{priv}}Acc(\tilde{f}(x,\tilde{\theta})), \tag{2}$$

where $\tilde{f}(x,\tilde{\theta})$ denotes the functionality of the attacking model.

Second, the defender aims to preserve the training performance of the logit-based FL protocol so that the perturbation scale should be bounded by a non-negative constant $\gamma$:

$$||p - p'||_b \leq \gamma, \tag{3}$$

where $p$ is the original logit on the public data, $p'$ is the corresponding perturbed logit, $\gamma$ is a pre-determined constant parameter and $||.||_b$ denotes the $L_b$ norm.

We note that the defender has no access to the adversary's model and may be even unaware that it is under attack since the attack happens at the server side. Therefore, the defender has to prevent the privacy risk from the semi-honest server during the whole training process.

**Defense Problem** Combining Equation (2) and (3), we can formulate a defense problem for the defender. However, this problem can not be directly solved since the attacking model parameters and its training details are unknown to the defender. Therefore, we need to approximate the first objective from the perspective of the defender.

The first step is to estimate the attacking model $\tilde{\theta}$. As the goal of $\tilde{\theta}$ is to approximate the defender's model $\theta$ in each training iteration, we estimate the attacking model obtained from the last iteration to be the same as the defender's model $\theta$ in the current iteration $T$:

$$\tilde{\theta}'_{T-1} = \theta_{T-1},$$

where $\tilde{\theta}'_{T-1}$ and $\theta_{T-1}$ are the estimated attacking model and the defender's model in the last training iteration respectively.

Without loss of generality, we assume that the adversary optimizes its attacking model through the gradient of an empirical loss on the public data, which is the most widely used optimization method in deep learning (Oliinyk, 2020). The gradient of an empirical loss with respect to parameter $\theta$ can be expressed as

$$G(\theta, p) = \nabla_\theta L_{emp}(f(x,\theta), p). \tag{4}$$

Based on the above assumptions, we restate the objectives of the defender as maximally changing the updated gradients of the estimated attacking model with minimum perturbation on the logit. That is, we can rewrite the first objective of the defender in iteration $T$ as maximizing the distance

between the gradients of the estimated attacking model updated through the original logit and the perturbed logit, in terms of $L_a$ norm:

$$\max_{p'}||G(\tilde{\theta}'_{T-1}, p') - G(\tilde{\theta}'_{T-1}, p)||_a$$
$$=\max_{p'}||G(\theta_{T-1}, p') - G(\theta_{T-1}, p)||_a \tag{5}$$

where $G(\theta, p)$ is the gradient of the empirical loss with respect to the parameters $\theta$.

Combining Equation (5) and (3), we therefore reformulate the defense problem as a constrained optimization problem:

$$\max_{p'}||G(\theta_{T-1}, p') - G(\theta_{T-1}, p)||_a \tag{6}$$
$$\text{s.t. } ||p - p'||_b \leq \gamma, \tag{7}$$

which allows the defender to trade off the utility and privacy in logit-based FL training. Worth mention that we can form multiple defense problems and corresponding defense strategies with different selections of $(a, b)$. In this paper, we set $(a, b) = (2, 1)$ and leave the other options as the future work.

**Defense solution** Deep learning models usually involves millions of parameters and thus solving Equation (6) s.t. Equation (7) with respect to each sample in the public dataset requires a large computational cost for clients, which is unaffordable for local devices in practice. Here, we give a simple heuristic solver to circumvent this computational issue. We perturb the logit $p$ on the public dataset in each dimension of itself by $Z$:

$$p' = p + Z \cdot e_j,$$

where $e_j$ denotes a one-hot vector with 1 in the $j$-th dimension of $p$ and 0's elsewhere. Then we select the one giving the largest perturbation in Equation (6). The local training with our defense in one iteration is given in Appendix B.2. We empirically show that this simple solution is effective in Section 5.3.

**Noise Selection and Privacy Guarantee** The added noise $Z$ has multiple choices, such as Laplace or Gaussian noise (Abadi et al., 2016). Following the prior works (Dwork et al., 2014), we adopt $Z$ as a Gaussian noise $Z = \mathcal{N}(0, \sigma^2)$, where $\sigma = \sqrt{\gamma}$ is a variance parameter. We show that our proposed perturbation based defense strategy in Algorithm 2 preserves $(\epsilon, \delta) - DP$ in Appendix C.

## 5 EXPERIMENT

In this section, we conduct experiments to answer two research questions: **RQ1:** Is the proposed AdaMSA effective against logit-based FL? **RQ2:** Is the proposed defense effective against AdaMSA, i.e. can it achieve a better utility and privacy trade-off?

### 5.1 EXPERIMENTAL SETUP

**Experimental Settings** To evaluate our proposed attack and defense strategy, we construct three experimental settings: Close-world, Open-world-CF and Open-world-TI for the image classification task. For Close-world, we utilize MNIST (LeCun et al., 1998) without labels as the unlabeled public dataset and EMNIST (Cohen et al., 2017) as the private dataset. For Open-world-CF, we utilize CIFAR10 (Krizhevsky et al., 2009) as the unlabeled public dataset and SVHN (Netzer et al., 2011) as the private dataset. For Open-world-TI, we adopt TinyImagenet (Le & Yang, 2015) as the unlabeled public dataset and SVHN (Netzer et al., 2011) as the private dataset. The details of experimental settings are given in Appendix E.1.

**Implementation** We adopt CNN as backbones for all clients' models. For the image classification problem, we choose the most commonly used cross-entropy loss as the empirical loss for both local and attacking models. We use a 2-layer CNN with (128,256) parameters as the attacking model and a CNN with same structure as the victim model to report the main result in Table 2. We provide details of hyperparameter choices in Appendix E.3. We repeat each experiment three times with random initialization and report the average accuracy.

**Baselines** We modify the conventional MSA method (Tramèr et al., 2016) in our setting as the attack baseline. For MSA baseline, we train an attacking model by approximating the victim's current logit in each round and report the highest accuracy of the attacking model during training.

| Setting | Defense | Victim Acc(%) | MSA(Tramèr et al., 2016) Acc(%) | AdaMSA Acc(%) |
|---------|---------|---------------|--------------------------------|---------------|
| Close-world | Unprotected (Li & Wang, 2019) | $83.43 \pm 1.07$ | $80.10 \pm 1.31$ | $\textbf{83.79} \pm 1.20$ |
| | Cross-domain (Lin et al., 2020) | $82.97 \pm 0.89$ | $79.43 \pm 1.07$ | $\textbf{82.68} \pm 1.01$ |
| | One-shot (Gong et al., 2021) | $68.23 \pm 1.21$ | $65.03 \pm 0.89$ | $\textbf{67.18} \pm 1.17$ |
| | DP-G (Sattler et al., 2021) | $74.68 \pm 2.15$ | $71.69 \pm 1.14$ | $\textbf{74.79} \pm 1.20$ |
| | DP-L (Gong et al., 2022) | $75.65 \pm 1.97$ | $72.01 \pm 1.21$ | $\textbf{75.53} \pm 1.09$ |
| Open-world-CF | Unprotected (Li & Wang, 2019) | $71.11 \pm 0.98$ | $68.77 \pm 1.13$ | $\textbf{71.25} \pm 0.71$ |
| | Cross-domain (Lin et al., 2020) | $53.57 \pm 1.13$ | $52.21 \pm 1.06$ | $\textbf{53.78} \pm 0.79$ |
| | One-shot (Gong et al., 2021) | $63.41 \pm 1.09$ | $60.03 \pm 0.98$ | $\textbf{62.49} \pm 0.91$ |
| | DP-G (Sattler et al., 2021) | $65.89 \pm 1.77$ | $62.20 \pm 1.39$ | $\textbf{65.61} \pm 1.31$ |
| | DP-L (Gong et al., 2022) | $65.22 \pm 1.96$ | $65.97 \pm 1.28$ | $\textbf{67.64} \pm 1.22$ |
| Open-world-TI | Unprotected (Li & Wang, 2019) | $71.11 \pm 0.98$ | $69.34 \pm 0.97$ | $\textbf{72.13} \pm 1.02$ |
| | Cross-domain (Lin et al., 2020) | $51.14 \pm 1.10$ | $51.17 \pm 1.05$ | $\textbf{53.54} \pm 0.80$ |
| | One-shot (Gong et al., 2021) | $63.09 \pm 1.14$ | $62.36 \pm 1.03$ | $\textbf{64.41} \pm 0.99$ |
| | DP-G (Sattler et al., 2021) | $65.76 \pm 2.01$ | $62.77 \pm 1.20$ | $\textbf{65.02} \pm 1.18$ |
| | DP-L (Gong et al., 2022) | $65.09 \pm 1.98$ | $66.11 \pm 1.13$ | $\textbf{68.41} \pm 1.04$ |

Table 2: Attack performance of our proposed AdaMSA and MSA baseline (Tramèr et al., 2016) on the victim model with various defense baselines in Close-world, Open-world-CF and Open-world-TI settings. Victim Acc denotes the performance of the victim model.

We compare our proposed defense strategy to several state-of-the-art baseline defenses: 1) *Unprotected (Li & Wang, 2019)*; 2) *Cross-domain (Lin et al., 2020)*; 3) *One-shot (Li et al., 2020)*; 4) *Differential Privacy with Gaussian (DP-G)* (Sattler et al., 2021) ; 5) *Differential Privacy with Laplacian (DP-L)* (Gong et al., 2022). The detailed explanation of the baselines is in Appendix E.2.

**Evaluation Metrics** For attack evaluation, we use the prediction accuracy of attacking model on the victim's private test dataset to measure the attack performance. For defense evaluation, we evaluate all defenses on a utility loss vs. privacy loss curve at various points of the defenses. The utility loss $\Delta U$ of the defense is defined as $\Delta U = U_d - U_0$, where $U_0$ is the accuracy of local model in unprotected baseline and $U_d$ is the accuracy of local model under different defense. The privacy loss is defined as the prediction accuracy of the attacking model on the defender's private test dataset.

## 5.2 ATTACK PERFORMANCE EVALUATION

**Main Results:** Table 2 shows the performance of our proposed AdaMSA and MSA baseline in three settings. We observe that AdaMSA achieves similar performance compared to the victim models in most of the defense baselines, indicating that AdaMSA can successfully steal the functionality of the victim's model. We also observe that attacking models of AdaMSA achieves up to 3.69% improvement in Close-world setting, 3.41% improvement in Open-world-CF setting and 2.79% in Open-world-TI setting compared to the baseline, indicating our attack design that combines historical logits generates a more informative attacking logit and thereby improve the attack performance.

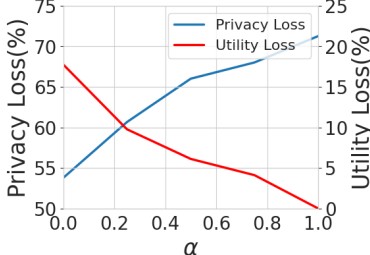 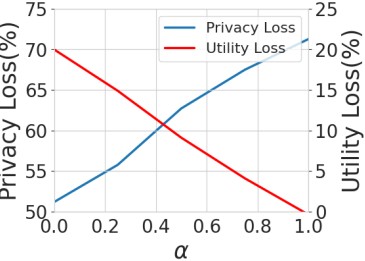

Figure 3: The relation between the utility loss/ privacy loss and $\alpha$ in Open-world-CF (left) and Open-world-TI (right) settings. The blue line denotes the utility loss-$\alpha$ curve and the red line denotes the privacy loss-$\alpha$ curve.

**Effect of Distance Between Private and Public Datasets** As mentioned previously in Section 3.3, distance between private and public datasets plays a crucial role in determining utility and privacy risk in logit-based FL. To evaluate the effect of this factor, we construct several mixed datasets in Open-world-CF and Open-world-TI setting as the public dataset through varying the weighting parameter $\alpha$. The results are reported in Figure 3. It can be observed that increasing the value of $\alpha$, i.e. decreasing the distance between the public and the private datasets, will increase the utility at

the cost of increasing privacy loss. This result indicates that the utility and privacy in logit-based FL are indeed two sides of a coin. That is, a more informative local logit results in a more informative ensemble logit to supervise the local model training, meanwhile it also exposes more privacy to the adversary. This result is consistent with our analysis in Section 3.3.

**Effect of Historical Logits** In order to evaluate the effect of our attack design which combines historical logits to generate a more informative attacking logit, we vary $T_0$ and test on the same victim model in two open-world settings. Note that, when $T_0 = 0$, it is equal to the situation that we perform MSA (Tramèr et al., 2016) baseline. From the results in Table 3, we find that, when $T_0$ increases from 0 to 4, the attack performance gradually increases by 2.59% in Open-world-CF setting and 3.10% in Open-world-TI setting, demonstrating that combining more historical logits indeed improves the attack performance. This is because

| $T_0$ | Attack Acc(%) | |
|---|---|---|
| | Open-world-CF | Open-world-TI |
| 0 | $68.77 \pm 1.13$ | $69.34 \pm 0.97$ |
| 1 | $70.01 \pm 0.73$ | $70.99 \pm 1.17$ |
| 2 | $70.47 \pm 1.03$ | $71.87 \pm 1.30$ |
| 3 | $71.25 \pm 0.71$ | $72.13 \pm 1.02$ |
| 4 | $\mathbf{71.36} \pm 0.88$ | $\mathbf{72.44} \pm 0.89$ |

Table 3: The effect of combining historical logits in the attacking logits on the attack performance of AdaMSA.

the historical predictions close to the current round can be viewed as the different views of the victim model on the public data. Therefore, our designed attacking logit can be benefited from the ensemble of these muti-view predictions, which improves model generalization ability on the test data (Mienye & Sun, 2022; Allen-Zhu & Li, 2020).

## 5.3 DEFENSE PERFORMANCE EVALUATION

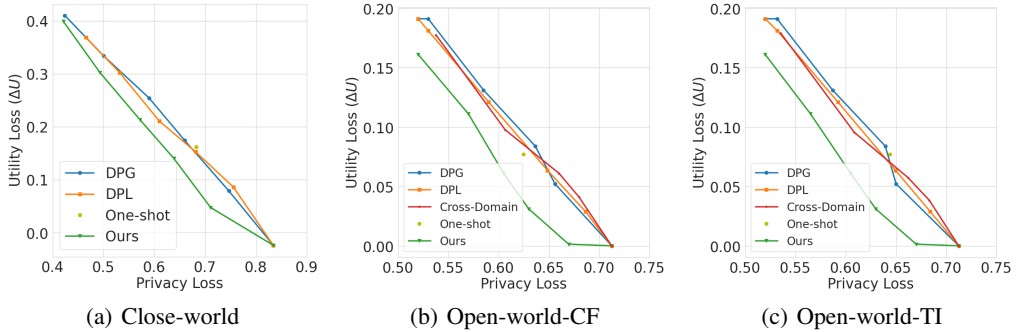

(a) Close-world      (b) Open-world-CF      (c) Open-world-TI

Figure 4: Defense performance evaluation in Close-world, Open-world-CF and Open-world-TI settings. The ideal trade-off curve resides on the bottom left corner in the figure.

The results of the state-of-the-art defense strategies in logit-based FL and our defense on three settings are reported in Figure 4. The X-axis represents the privacy loss, i.e. the capability for adversarial to infer a client's private model. Y-axis represents the utility loss brought by the defense methods. Comparing to the state-of-the-art baselines, our proposed defense is closest to the ideal trade-off, which should reside in the bottom left corner in Figure 4. For example, when privacy loss is 0.7, the utility loss of our defense is around 8% less than DP-G and DP-L in Close-world setting. This result indicates that our defense can provide a better utility and privacy trade-off compared to the state-of-the-art defense baselines.

## 6 CONCLUSION

In this paper, we provide the first theoretical and empirical analysis of a hidden privacy risk in logit-based FL that the semi-honest server can infer clients' private models according to logits. To quantify the impacts of the privacy risk, we develop an effective attack named Adaptive Model Stealing Attack (AdaMSA) by leveraging historical logits during training and provide a theoretical analysis on the bound of the privacy risk. Moreover, we propose a perturbation-based defense that perturbs the transmitted logit in the direction that minimizes the privacy risk while maximally preserving the training performance. The empirical results on three experimental settings demonstrate the effectiveness of our proposed attack and defense.

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

## A  RELATED WORK

We summarize the existing privacy attacks against FL in Table 4. These efforts are all white-box attacks that have strong assumptions (e.g. the adversary knows the local model architecture and detailed training information such as gradients).

| Attack | Adversary | Adversary's Goal | Adversary's Knowledge | | | Applicable for |
|---|---|---|---|---|---|---|
| | | | Model | Gradient | Logit | Logit-based FL |
| CPA (Nasr et al., 2019) | Semi-honest/ Malicious Client | Infer Membership | ✓ | ✓ | ✗ | ✗ |
| mGAN (Wang et al., 2019) | Malicious Client | Infer Class Representative | ✓ | ✓ | ✗ | ✗ |
| UFL (Melis et al., 2019) | Semi-honest/ Malicious Client | Infer Property | ✓ | ✓ | ✗ | ✗ |
| DLG (Zhu et al., 2019) | Semi-honest Server | Infer Training Data | ✓ | ✓ | ✓ | ✗ |
| InvertGrad (Geiping et al., 2020) | Semi-honest Server | Infer Training Data | ✓ | ✓ | ✓ | ✗ |
| AdaMSA (ours) | Semi-honest Server | Infer Model | ✗ | ✗ | ✓ | ✓ |

Table 4: Existing privacy attacks against FL.

## B  ALGORITHM

### B.1  ALGORITHM OF AdaMSA

In Algorithm 1, we provide the algorithm of AdaMSA in one training iteration described in Section 3.2.

---
**Algorithm 1:** AdaMSA in one iteration

---
**Input:** Public dataset $\mathcal{D}_{pub}$, local logits $\{p_t, t \in [1, T]\}$ predicted on $\mathcal{D}_{pub}$, attacking model $\tilde{\theta}$, iteration number $T$.
**Output:** Updated attacking model $\tilde{\theta}$.
1: **while** not converge **do**
2:     Randomly shuffle $\mathcal{D}_{pub}$
3:     **for** i in range($|\mathcal{D}_{pub}|$) **do**
4:         $\hat{p}_T \leftarrow \sum_{t \in [T-T_0, T]} w_0 \cdot \frac{t}{T} \cdot p_t$
5:         $\tilde{L} \leftarrow L_{emp}(\hat{p}_T, \tilde{p}_T)$
6:         $\tilde{\theta} \leftarrow \tilde{\theta} - \nabla_{\tilde{\theta}} \tilde{L}$
7:     **end for**
8: **end while**
9: **return** $\tilde{\theta}$

---

### B.2  ALGORITHM OF OUR PROPOSED DEFENSE

In Algorithm 2, we provide the algorithm of our proposed defense described in Section 4. We highlight our defense design in the blue lines.

---

**Algorithm 2:** Local Training with defense

---

**Input:** Public dataset $\mathcal{D}_{pub}$, local dataset $\mathcal{D}_{priv} : \{x, y\}$, local model in last iteration $\theta_{T-1}$,
    variance parameter $\sigma$.
**Output:** Updated local model $\theta_T$.
 1: **Local Training:**
 2: Train local model with $\mathcal{D}_{priv}$ and update $\theta$
 3: **Logit Ensemble:**
 4: **for** $x_i$ in $\mathcal{D}_{pub}$ **do**
 5:     $p_i \leftarrow \theta_T(x_i)$
 6: **end for**
 7: **for** each dimension $e_j$ of $p_i$ **do**
 8:     $p_i' \leftarrow p_i + \mathcal{N}(0, \sigma^2) \cdot e_j$, for $x_i \in \mathcal{D}_{pub}$
 9:     Calculate $\sum\limits_{x_i \in \mathcal{D}_{pub}} ||G_i(\theta_{T-1}, p_i') - G_i(\theta_{T-1}, p_i)||_2$ according to Equation (4)
10: **end for**
11: $p' \leftarrow \text{argmax}_{p'} \sum\limits_{x_i \in \mathcal{D}_{pub}} ||G_i(\theta_{T-1}, p_i) - G_i(\theta_{T-1}, p_i')||_2$
12: Upload $p'$ to the server and then obtain the corresponding ensemble logits from the server
13: **Distillation:**
14: Train local model with ensemble logits on $\mathcal{D}_{pub}$ and update $\theta$
15: **return** $\theta_T$

---

## C   PRIVACY GUARANTEE OF DEFENSE

We first give definition of differential privacy and $l_2$-sensitivity (Dwork et al., 2014; 2006). Then we show that our proposed perturbation based defense strategy in Algorithm 2 preserves $(\epsilon, \delta) - DP$.

**Definition 3.** (Differential Privacy). A randomized mechanism $f : \mathcal{X} \rightarrow \mathcal{Y}$ is $(\epsilon, \delta)$-DP, if and only if for every pair of datasets $X, X' \in \mathcal{X}$ that only differ in one sample and every possible output $E \subseteq range(f)$, the following inequality holds:

$$\mathbb{P}[f(X) \in E] \leq e^{\varepsilon} \mathbb{P}[f(X') \in E] + \delta.$$

where $\epsilon > 0$ represents the privacy budget, $\delta > 0$ represents the probability that the maximum distance is not bounded by $\epsilon$ and $range(f)$ denotes the set of all possible outputs of $f$.

**Definition 4.** ($l_2-$sensitivity). The $l_2$-sensitivity of a function $f : \mathcal{X} \rightarrow \mathbb{R}^d$ is defined as

$$\Delta_2(f) = \max_{X, X' \in \mathcal{X}} ||f(X) - f(X')||_2.$$

**Theorem 3.** For any $\epsilon > 0$ and $\delta \in (0, 1)$, the mechanism described in Algorithm 2 with a sensitivity $\Delta_2$ preserves $(\epsilon, \delta) - DP$ if and only if $\sigma \geq \sqrt{2ln1.25\delta} \cdot \frac{\Delta_2}{\epsilon}$.

We provide the proof of Theorem 3 in Appendix D.3.

# D PROOF

## D.1 PROOF OF THEOREM 1

*Proof.* According to the definition given in Equation (1), we have

$$
\begin{aligned}
&\epsilon_{\mathcal{D}_{mix}}(\theta, f_p) \\
=&\mathbb{E}_{x\sim\mathcal{D}_{mix}}[|\theta(x) - f_p(x)|] \\
=&\frac{1}{|\mathcal{D}_{mix}|}\sum_{x_i\in\mathcal{D}_{mix}}|\theta(x_i) - f_p(x_i)| \\
=&\frac{1}{|\mathcal{D}_{mix}|}[\sum_{x_i\in\mathcal{D}_{priv}}|\theta(x_i)-f_p(x_i)| + \sum_{x_j\in\mathcal{D}_{pub}}|\theta(x_j)-f_p(x_j)|] \\
=&\alpha\cdot\frac{1}{|\alpha\mathcal{D}_{mix}|}\sum_{x_i\in\mathcal{D}_{priv}}|\theta(x_i) - f_p(x_i)|+ \\
&(1-\alpha)\cdot\frac{1}{|(1-\alpha)\mathcal{D}_{mix}|}\sum_{x_j\in\mathcal{D}_{pub}}|\theta(x_j) - f_p(x_j)| \\
=&\alpha\epsilon_{\mathcal{D}_{priv}}(\theta) + (1-\alpha)\epsilon_{\mathcal{D}_{pub}}(\theta).
\end{aligned}
$$

## D.2 PROOF OF THEOREM 2

*Proof.* The proof of Theorem 2 builds on the restatement of the main theorem in (Ben-David et al., 2006) by (Blitzer et al., 2007), which demonstrates that for any hypotheses $h_1, h_2 \in \mathcal{H}$,

$$|\epsilon_{\mathcal{D}_{priv}}(h_1, h_2) - \epsilon_{\mathcal{D}_{pub}}(h_1, h_2)| \leq \frac{1}{2}d_{\mathcal{H}\Delta\mathcal{H}}(\mathcal{D}_{priv}, \mathcal{D}_{pub}), \tag{8}$$

and the triangle inequality for classification error (Crammer et al., 2008), which demonstrates that for any hypothesis $h_1, h_2, h_3 \in \mathcal{H}$ with respect to $\mathcal{D}$,

$$\epsilon_{\mathcal{D}}(h_1, h_2) \leq \epsilon_{\mathcal{D}}(h_1, h_3) + \epsilon_{\mathcal{D}}(h_2, h_3). \tag{9}$$

Then we can derive the bound of the difference between the empirical risk of the mixed dataset $\mathcal{D}_{mix}$ and the private dataset $\mathcal{D}_{priv}$ as

$$
\begin{aligned}
&|\epsilon_{\mathcal{D}_{mix}}(h, f_p) - \epsilon_{\mathcal{D}_{priv}}(h, f_p)| \\
=&(1-\alpha)|\epsilon_{\mathcal{D}_{priv}}(h, f_p) - \epsilon_{\mathcal{D}_{pub}}(h, f_p)| \\
=&(1-\alpha)\{|[\epsilon_{\mathcal{D}_{priv}}(h, f_p)-\epsilon_{\mathcal{D}_{priv}}(h, h^*)]+[\epsilon_{\mathcal{D}_{pub}}(h, h^*)- \\
&\epsilon_{\mathcal{D}_{pub}}(h, f_p)] + [\epsilon_{\mathcal{D}_{priv}}(h, h^*) - \epsilon_{\mathcal{D}_{pub}}(h, h^*)]|\} \\
\leq&(1-\alpha)[|\epsilon_{\mathcal{D}_{priv}}(h, f_p) - \epsilon_{\mathcal{D}_{priv}}(h, h^*)|+|\epsilon_{\mathcal{D}_{pub}}(h, f_p)- \\
&\epsilon_{\mathcal{D}_{pub}}(h, h^*)| + |\epsilon_{\mathcal{D}_{priv}}(h, h^*) - \epsilon_{\mathcal{D}_{pub}}(h, h^*)|] & (10) \\
\leq&(1-\alpha)[|\epsilon_{\mathcal{D}_{priv}}(h^*, f_p)| + |\epsilon_{\mathcal{D}_{pub}}(h^*, f_p)|+ \\
&|\epsilon_{\mathcal{D}_{priv}}(h, h^*) - \epsilon_{\mathcal{D}_{pub}}(h, h^*)|] & (11) \\
\leq&(1-\alpha)(\frac{1}{2}d_{\mathcal{H}\Delta\mathcal{H}}(\mathcal{D}_{priv}, \mathcal{D}_{pub}) + \lambda). & (12)
\end{aligned}
$$

In the proof, Equation (10) is derived from the absolute value inequality, Equation (11) is derived from the triangle inequality introduced in Equation (9) and the last step is derived by substituting Equation (8) and $\lambda = \epsilon_{\mathcal{D}_{priv}}(h^*) + \epsilon_{\mathcal{D}_{pub}}(h^*)$.

## D.3 PROOF OF THEOREM 3

*Proof.* The proof of Theorem 3 is based on the definition of Gaussian differential privacy and composition theorem of DP algorithms (Dwork et al., 2014).

**Theorem 4.** (Gaussian Differential Privacy). Let $\epsilon \in (0, 1)$ be arbitrary. For $c^2 > 2ln(1.25/\delta)$, the Gaussian Mechanism with parameter $\sigma \geq c\Delta_2(f)$ is $(\epsilon, \delta)$-differentially private.

**Theorem 5.** (Composition of DP Algorithms). Suppose $M = (M_1, M_2, ..., M_k)$ is a sequence of algorithms, where $M_i$ is $(\epsilon_i, \delta_i)$-DP, and the $M_i$'s are potentially chosen sequentially and adaptively. Then $M$ is $(\sum_{i=1}^{k} \epsilon, \sum_{i=1}^{k} \delta)$-DP.

According to Theorem 4, when fixing privacy budget $\epsilon_i$ and $\delta_i$, we can calibrate the added noise with proper $\sigma$ according to the perturbation constraint scale $\gamma$ for each client. The magnitude of $\sigma$ remains unchanged and thus the perturbed output of each client still preserves $(\epsilon_i, \delta_i) - DP$. Then, according to Theorem 5, the composition property of DP algorithms ensure that the ensemble result computed using the perturbed outputs is still $(\sum_{i=1}^{k} \epsilon, \sum_{i=1}^{k} \delta)$ private. Therefore, our proposed defense strategy in Algorithm 2 preserves $(\epsilon, \delta) - DP$.

## E   EXPERIMENT

### E.1   DETAILS OF EXPERIMENT SETTINGS

- **Close-world:** Following the same experimental setting in Fedmd (Li & Wang, 2019), we first construct a simple Close-world setting that public data and private data are similar. We use MNIST (LeCun et al., 1998) without labels as the unlabeled public dataset and EMNIST (Cohen et al., 2017) as the private dataset. We randomly select 300 images (30 for each class) in EMNIST as the private training dataset and then identically distributed to 10 clients. The rest of the private dataset (EMNIST) is used as the private test dataset.

- **Open-world-CF:** To construct a cross-domain experimental setting, we first discard the labels in CIFAR10 (Krizhevsky et al., 2009) and utilize it as the unlabeled public dataset. Then we randomly select 1000 images in each class of SVHN as the private training dataset (Netzer et al., 2011) and identically distributed to 10 clients. The rest of the private dataset (SVHN) is used as the private test dataset.

- **Open-world-TI:** Similar to Open-world-CF, we select the first 20 classes in TinyImagenet (Le & Yang, 2015) with 500 images per class without labels as the unlabeled public dataset and SVHN (Netzer et al., 2011) as the private dataset. Next we randomly select 1000 images in each class of SVHN as the private training dataset and identically distributed to 10 clients. The rest of the private dataset (SVHN) is used as the private test dataset.

### E.2   DETAILS OF BASELINES

- **Unprotected (Li & Wang, 2019)**: In this baseline, we follow the training process of the general logit-based FL approach (Li & Wang, 2019) and use public data that is similar to the private data. No attempt is made at defense.

- **Cross-domain (Lin et al., 2020)**: In this baseline, the public dataset is selected from another irrelevant and insensitive domain to prevent privacy leakage (i.e. the mixed public dataset when $\alpha = 0$).

- **One-shot (Li et al., 2020)**: In this baseline, we utilize one-shot distillation on unlabeled and domain-robust public data. Specifically, clients only communicate once with server.

- **Differential Privacy (DP)**: Recent works (Gong et al., 2022; Sattler et al., 2021) adopt Gaussian and Laplacian DP and add noise to the transmitted updates. We conduct their strategies on the unprotected baseline as **DP-G** (Sattler et al., 2021) and **DP-L** baseline (Gong et al., 2022), respectively.

### E.3   DETAILS OF IMPLEMENTATION

For a fair comparison, we set the perturbation scale $\gamma$ to be the same as 0.01 for DP-G, DP-L and our defense in Table 3 & 6 and $T_0, w_0$ is set to be 3 and 0.5 respectively. All models are optimized by Adam with a 0.01 learning rate. For Close-world setting, the batch size of local training on private data is 10 and on public data is 128. The number of communication rounds is 10. For Open-world settings, the batch size of local training on the private dataset is 32 and on the public dataset is 256. The number of communication rounds is 30.

### E.4   EFFECT OF VICTIM MODEL HETEROGENEITY

We vary the victim model parameters and architectures in the Open-world-CF setting to evaluate the effect of victim model heterogeneity on the attack performance. We also find a similar result in the Open-world-TI setting, which is omitted due to the space limit. For a fair comparison, we perform the experiments in the unprotected baseline. The victim models vary from CNN to Alexnet (Krizhevsky et al., 2017) with different hyperparameters. The details are reported in Table 5. The results show that AdaMSA can achieve similar accuracy of the victim models under different model hyperparameters. This indicates that AdaMSA is robust to heterogeneous victim models in our setting.

| Model | Layer | Parameter | Victim Acc(%) | Attack Acc(%) |
|---|---|---|---|---|
| CNN | 2 | 128,256 | $71.11 \pm 1.21$ | $69.33 \pm 0.99$ |
| CNN | 2 | 128,512 | $70.15 \pm 1.14$ | $70.28 \pm 1.02$ |
| CNN | 2 | 256,512 | $70.56 \pm 1.30$ | $70.42 \pm 0.79$ |
| CNN | 3 | 64,128,256 | $71.11 \pm 0.98$ | $71.25 \pm 0.71$ |
| CNN | 3 | 128,192,256 | $69.64 \pm 1.28$ | $69.19 \pm 1.13$ |
| Alexnet (Krizhevsky et al., 2017) | 8 | - | $75.10 \pm 1.29$ | $75.33 \pm 1.24$ |

Table 5: The effect of victim model heterogeneity on the attack performance of AdaMSA in Open-world-CF setting.

## F   FUTURE DIRECTION

Future directions include analyzing the impact of different selection for $(a, b)$ in Equation (8) and (9) as the defense strategies and ensuring higher level of privacy through encryption techniques such as homomorphic encryption (Rivest et al., 1978) or secure multiparty computation (Yao, 1982). Another interesting future direction is to investigate other hidden privacy risks in logit-based FL, including inferring training data membership and directly reconstructing the training input.

