# OpenReview forum: "Quantifying and Defending against the Privacy Risk in Logit-based Federated Learning"
_ICLR.cc/2024/Conference — ICLR 2024 Conference Withdrawn Submission_

### Official Review · Reviewer_K1GZ · 2023-10-31

**Soundness:** 2 fair
**Presentation:** 3 good
**Contribution:** 2 fair
**Rating:** 5
**Confidence:** 4

**Summary:**

This paper explores logit-based Federated Learning (FL) methods that aim to protect data privacy. It highlights a previously unnoticed privacy risk where a semi-honest server could potentially learn clients' private models from shared logits. The paper introduces an attack called Adaptive Model Stealing Attack (AdaMSA) and proposes a defense strategy to mitigate this risk. Experimental results confirm the effectiveness of the attack and defense strategy.

**Strengths:**

+ A novel Adaptive Model Stealing Attack (AdaMSA) is proposed to quantify the privacy risk of logit-based FL.
+ A simple yet effective defense strategy is proposed to achieve better privacy-utility trade-off.
+ A bounded analysis of privacy risks is provided for the proposed privacy attacks.
+ Extensive case studies.

**Weaknesses:**

- The value of the research question requires further justification.
 - The outcomes of the experiment need to be made more convincing.
- Limited in-depth comparison with state-of-the-art solutions.

**Questions:**

Q1: The motivation of this article requires further justification by providing additional evidence. The authors mentioned that logit-based FL was developed to achieve communication-efficient FL. However, this is not a mainstream FL framework nor a mainstream communication-efficient FL framework. For example, asynchronous FL, gradient compression-based FL, gradient quantization-based FL, and generative learning-based one-shot FL are all widely adopted. Therefore, the reviewer's first concern is whether it is necessary and valuable to analyze the privacy risks of logit-based FL.

Q2: The objectives of the adversary's attack warrant further examination. While the paper articulates the adversary's aim as acquiring the private model θ, it is imperative to delve deeper into whether this private model θ can be subsequently leveraged for malicious purposes. It is worth noting that previous works in the field of privacy attacks primarily focus on the exfiltration of a client's confidential data. Consequently, a critical concern arises: can the adversary utilize the private model θ to reverse-engineer the original training data? This crucial aspect of the adversary's capabilities necessitates thorough investigation and discussion to assess the potential privacy risks associated with the acquired private model.

Q3: More advanced baselines need to be included to highlight the superiority of the proposed privacy attacks. Considering that FL was proposed in 2016 and the baseline scheme compared in this article was also proposed in 2016, whether this scheme is representative still needs to be discussed. It would be better if the authors could consider more baseline solutions (such as [1]).

[1] Takahashi H, Liu J, Liu Y. Breaching FedMD: Image Recovery via Paired-Logits Inversion Attack[C]//Proceedings of the IEEE/CVF Conference on Computer Vision and Pattern Recognition. 2023: 12198-12207.

Q4: There is merit in exploring additional, perhaps more straightforward, security mechanisms to corroborate and strengthen the privacy assurances of logit-based Federated Learning (FL). It is essential to recognize that the fundamental premise underpinning the attacks in this article hinges on the server's ability to access the logits uploaded by the client. However, it is possible to mitigate this vulnerability through the implementation of secure aggregation techniques and the utilization of hardware-based Trusted Execution Environments (TEEs). These measures can effectively safeguard against the server's unauthorized access to logits. It is pertinent to acknowledge that these considerations do not diminish the innovative contributions of the article. Nevertheless, it would be advantageous for the authors to engage in a discourse on these potential defense mechanisms to provide a more comprehensive understanding of the robustness of logit-based FL with respect to privacy concerns.

---

> ### Author Response · Authors · 2023-11-21
> **Response to Weakness 1-3 and Question 1-3**
>
> **Weakness 1 and Question 1:**
> >The value of the research question requires further justification.
> >The motivation of this article requires further justification by providing additional evidence. The authors mentioned that logit-based FL was developed to achieve communication-efficient FL. However, this is not a mainstream FL framework nor a mainstream communication-efficient FL framework. For example, asynchronous FL, gradient compression-based FL, gradient quantization-based FL, and generative learning-based one-shot FL are all widely adopted. Therefore, the reviewer's first concern is whether it is necessary and valuable to analyze the privacy risks of logit-based FL.
>
> **Response:**
> Thank you for your concerns regarding the significance of logit-based FL and our work in the community of FL. we address these concerns in the *General Response 1 and 2*.
>
> **Weakness 2 and Question 2:**
> >The outcomes of the experiment need to be made more convincing.
> >The objectives of the adversary's attack warrant further examination. While the paper articulates the adversary's aim as acquiring the private model $\theta$, it is imperative to delve deeper into whether this private model $\theta$ can be subsequently leveraged for malicious purposes. It is worth noting that previous works in the field of privacy attacks primarily focus on the exfiltration of a client's confidential data. Consequently, a critical concern arises: can the adversary utilize the private model $\theta$ to reverse-engineer the original training data? This crucial aspect of the adversary's capabilities necessitates thorough investigation and discussion to assess the potential privacy risks associated with the acquired private model.
>
> **Response:**
> We appreciate the reviewer‘s thoughtful consideration of the adversary’s objectives, particularly regarding the potential malicious use of the acquired private model $\theta$.
> In our initial exploration of the potential privacy risks in logit-based FL, our primary emphasis is on treating the private model itself as the privacy element and our focus centers on the identification and quantification of risks associated with the private model. In both the introduction and Appendix F, we have discussed this potential direction as a promising avenue for future work.
> We also want to underscore that compromising the privacy of logit-based FL is nontrivial, given that the only knowledge available to a semi-honest adversary is the victim's logits on the public dataset. This limitation is a critical aspect of our study, and we appreciate the opportunity to clarify our current focus and future directions.
>
> **Weakness 3 and Question 3:**
> >Limited in-depth comparison with state-of-the-art solutions.
> >More advanced baselines need to be included to highlight the superiority of the proposed privacy attacks. Considering that FL was proposed in 2016 and the baseline scheme compared in this article was also proposed in 2016, whether this scheme is representative still needs to be discussed. It would be better if the authors could consider more baseline solutions (such as [1]).
>
> **Response:**
> We appreciate the reviewer‘s valuable suggestion to incorporate more advanced baselines in order to highlight the superiority of our proposed privacy attacks. In the context of logit-based FL, we do recognize the presence of numerous variants (as explained in *General Response 2*). However, it's noteworthy that these variants can be seen as combinations of parameter-based and logit-based FL, thereby leading to privacy issues similar to those encountered in parameter-based FL.
>
> Regarding the comparison with the baseline solution mentioned in [1], we are thankful to the reviewer for bringing our attention to this contemporary work. It is important to clarify a key difference between our study and [1]. The tasks undertaken in the two works differ significantly. [1] focuses on reconstructing class representation of private data, which is comparatively much easier than recovering the original private data. As also highlighted in the Limitation section of [1], the attacker cannot directly obtain anything calculated from the private dataset, and reconstructing the exact image remains challenging in the context of logit-based FL. Moreover, we provide both theoretical and empirical analysis of the privacy risk in logit-based FL.
>
> It is essential to recognize that our study and [1] are oriented towards different directions in exploring the privacy risks of logit-based FL. We believe that our work and [1] can be viewed as complementary efforts, as they contribute to an empirical understanding of privacy risks in logit-based FL that recovers private class representatives. A more effective attacking model, as proposed in our work, can be instrumental in reversing private class representations, thereby advancing the progression towards recovering the original private data.

---

> > ### Author Response · Authors · 2023-11-21
> > **Response to Question 4**
> >
> > **Question 4:**
> > >There is merit in exploring additional, perhaps more straightforward, security mechanisms to corroborate and strengthen the privacy assurances of logit-based Federated Learning (FL). It is essential to recognize that the fundamental premise underpinning the attacks in this article hinges on the server's ability to access the logits uploaded by the client. However, it is possible to mitigate this vulnerability through the implementation of secure aggregation techniques and the utilization of hardware-based Trusted Execution Environments (TEEs). These measures can effectively safeguard against the server's unauthorized access to logits. It is pertinent to acknowledge that these considerations do not diminish the innovative contributions of the article. Nevertheless, it would be advantageous for the authors to engage in a discourse on these potential defense mechanisms to provide a more comprehensive understanding of the robustness of logit-based FL with respect to privacy concerns.
> >
> > **Response:**
> > We appreciate the reviewer's insightful suggestion to explore additional security mechanisms that can bolster the privacy assurances of logit-based FL.
> >
> > While we acknowledge the effectiveness of security mechanisms such as TEE, secure multi-party sharing, and Homomorphic Encryption (HE) in enhancing privacy protection, it is essential to note that they also come with limitations. These mechanisms may demand significant additional computational or storage resources for the server.
> >
> > In our paper, our primary focus is on privacy attack and defense. However, we acknowledge the importance of considering broader aspects of security, including potential security attacks such as poisoning attacks and security mechanisms. To provide a more comprehensive understanding, we plan to include a discussion on these topics in the appendix of our revised manuscript. This additional content will contribute to a more holistic exploration of privacy and security considerations in logit-based Federated Learning.

---

> > > ### Comment · Reviewer_K1GZ · 2023-11-21
> > > **Response to Authors**
> > >
> > > Thanks to the authors for their detailed and timely responses! First, the authors have addressed some of my concerns, but not quite enough. I still have the following concerns that need to be further addressed:
> > >
> > > - Whether the Logit-based FL framework is representative is unclear. Although the authors cited literature [6–8] to support their claims, literature [6–8] is neither a mature industrial application nor a classic academic work. I remain concerned about the practicality of this article's research questions.
> > >
> > > - Regarding Q4, I think it is very common in industrial applications to sacrifice additional computing and storage resources to fully ensure user privacy. If the security and privacy attacks explored in this article can be avoided using TEEs, the authors would be wise to discuss and provide more insights into why the proposed attacks are not evaluated in the context of these security mechanisms.
> > >
> > > I look forward to responses from the authors!

---

> > > > ### Author Response · Authors · 2023-11-21
> > > > **Response to Concern 2**
> > > >
> > > > **Concern 2: Why we do not take security mechanisms such as TEEs as defense?**
> > > >
> > > > **Response:**
> > > > While we acknowledge the effectiveness of security mechanisms such as Trusted Execution Environments (TEEs) in defending against privacy attacks, it's essential to consider their limitations. FL is as a distributed framework involving hundreds or even thousands of participants (e.g., phones or vehicles) in industrial applications. Implementing a uniform TEE for all these diverse devices poses significant challenges and may not be a practical solution.
> > > >
> > > > We appreciate the reviewer's insightful suggestion to explore additional security mechanisms as defense approaches and we will add the corresponding discussions in the Appendix.

---

> > > > > ### Comment · Reviewer_K1GZ · 2023-11-22
> > > > > **Response to Authors**
> > > > >
> > > > > Thanks to the authors for their replies! In fact, defense measures such as TEE, security aggregation mechanism, and DP are practical and secure solutions whether they involve thousands of cross-device FL scenarios or cross-silo FL scenarios involving a few organizations.

---

> ### Author Response · Authors · 2023-11-21
> **Response to Concerns 1**
>
> **Concern 1: Why we need logit-based FL?**
>
> **Response:**
> The references [6-8] cited in the previous response primarily emphasize potential applications of logit-based FL. For a deeper understanding of the classic academic framework of logit-based FL, we recommend exploring two pioneering works: [1] (proposed in 2019, citation 565) and [2] (proposed in 2020, citation 596). It's crucial to recognize that logit-based FL is a relatively new framework, having emerged within the last four years. Despite its early stage of development, it demonstrates promise as an evolving research direction, as seen in the increasing number of papers related to "Federated Learning" and "distillation" on arXiv in recent years:
>
> | | 2019 | 2020 | 2021 | 2022 | 2023|
> | :-:| :-:| :-:| :-:|:-: |:-: |
> |**Number of Paper** |5 | 22 | 15 | 57 | 82 |
>
> The growth is propelled by the framework's potential to address several crucial challenges in FL, including efficiency, privacy, and model heterogeneity.
>
> It's noteworthy that while parameter-based FL serves as a standard and well-established norm for cross-device federated learning, it encounters limitations in many real-world scenarios that demand model heterogeneity and high efficiency (e.g., edge computing, Internet of Vehicles, etc.). Furthermore, recent studies [3-4] have unveiled the possibility of reconstructing the original input from the shared parameters, thereby compromising the most important property of federated learning: privacy. This underscores the significance of exploring alternative frameworks like logit-based FL to better adapt to the complexities of diverse applications.
>
> **Reference:**
> [1] Li, Daliang, and Junpu Wang. "Fedmd: Heterogeneous federated learning via model distillation." arXiv preprint arXiv:1910.03581 (2019).
> [2] Lin, Tao, et al. "Ensemble distillation for robust model fusion in federated learning." Advances in Neural Information Processing Systems 33 (2020): 2351-2363.
> [3] Zhu, Ligeng, Zhijian Liu, and Song Han. "Deep leakage from gradients." Advances in neural information processing systems 32 (2019).
> [4] Zhao, Bo, Konda Reddy Mopuri, and Hakan Bilen. "idlg: Improved deep leakage from gradients." arXiv preprint arXiv:2001.02610 (2020).

---

> > ### Comment · Reviewer_K1GZ · 2023-11-22
> > **Response to Authors**
> >
> > Thanks to the authors for their detailed and timely responses! Indeed, I acknowledge that there are some excellent works in the research community focusing on logit-based FL frameworks, however, this does not mean that the privacy issues of this type of framework are worthy of attention. Consistent with the concerns of other reviewers, the need for research on privacy issues in logit-based FL frameworks still seems to lack sufficient evidence to support it.

---

### Official Review · Reviewer_sfwc · 2023-10-31

**Soundness:** 3 good
**Presentation:** 3 good
**Contribution:** 2 fair
**Rating:** 3
**Confidence:** 5

**Summary:**

This work develops a model stealing attack (AdaMSA) in logit-based federated learning. Additionally, it provides a theoretical analysis of the bounds of privacy risks. It also proposes a simple but effective defense strategy that perturbs the transmitted logits in the direction that
minimizes the privacy risk while maximally preserving the training performance.

**Strengths:**

- The proposed attack is effective with a high attack success rate.
- Theoretical analysis is conducted to quantify the privacy risks in logit-based FL.
- Extensive empirical results support the theoretical analysis

**Weaknesses:**

- The proposed method is impractical to be executed or evaluated in real-world scenarios.
- The privacy risk metric can not express the true privacy risk of the setting.
- The notations are vague which makes it very hard to follow the analysis of the work

**Questions:**

1. Why does the accuracy on the private dataset express the success rate of the model stealing attack? What if the attack mode is very generalized which achieves high accuracy in the same data distribution?

2.  Since $D_{pub}$ is unlabeled, how to quantify Eq. 1 for $D_{mix}$ ?

3. The construction of $D_{mix}$ is based on the empirical sets of $D_{priv}$ and $D_{pub}$. How does it reflect the true distribution of $D_{priv}$ and $D_{pub}$?

4. Since the adversary cannot touch $D_{priv}$, how to construct $D_{mix}$?

5. What is the difference between the theoretical analysis of the work compared to Blitzer et al., 2007?

---

> ### Author Response · Authors · 2023-11-20
> **Response to Weakness 1-3**
>
> **Weakness 1-2:**
> > The proposed method is impractical to be executed or evaluated in real-world scenarios.
> The privacy risk metric can not express the true privacy risk of the setting.
>
> **Response:**
> If we understand correctly, the concern raised about the impracticality of our proposed method revolves around two key aspects: 1) **the feasibility and practicality of constructing the mix dataset as the public dataset**, and 2) **the evaluation of the attacking model on the private test data as the privacy metric**. We appreciate the opportunity to address these concerns and provide additional context on the motivation behind our approach.
>
> - **The feasibility and practicality of constructing the mix dataset as the public dataset:**
>
> In logit-based FL, the public dataset can either have the same distribution as the private dataset or a different distribution in an irrelevant domain. Recognizing that the correlation (i.e., distance) between the private and public datasets is crucial in determining the performance of logit-based FL methods, we want to first quantify and vary this correlation.
>
> To the best of our knowledge, few studies provide methods that can continuously vary the distance between two datasets. Therefore, we propose to construct a mix dataset as demonstrated in Fig. 2(a). **We want to emphasis that, in our setting, the mix dataset is used as the public dataset in the standard logit-based FL setting [1].**
>
> Constructing the mix dataset allows us to systematically manipulate the correlation between the mix and private datasets, providing a controlled environment to evaluate the privacy risks in logit-based FL. While we acknowledge the complexity of constructing such a mix dataset, we believe it is a necessary step to better understand and quantify the privacy implications in logit-based FL.
>
> - **The evaluation of the attacking model on the private test data as the privacy metric:**
>
> We would like to emphasize two critical points. Firstly, **the evaluation of the attacking model's performance occurs after the training phase**, ensuring that the adversary (i.e., server) does not have access to the private data during the model training process. The server remains unaware of the specifics of the private data throughout the training phase, which is also the exact phase that the attack and defense take place.
>
> Secondly, in our study, we identify the local model itself as the privacy element for the client that the adversary aims to attack and defend. To assess the success of the attack, we employ a widely adopted metric in model stealing attacks, which is **the functionality of the model** [2-4]. Specifically, we evaluate whether the attacking model can achieve high classification accuracy on the defender’s classification task. This choice allows us to gauge the effectiveness of the attack in terms of replicating the functionality of the private model.
>
> **Weakness 3:**
> >The notations are vague which makes it very hard to follow the analysis of the work.
>
> **Response:**
> We appreciate the reviewer's feedback regarding the vagueness of notations in our work, and we are committed to improving the clarity of our manuscript.
>
> To address this concern effectively, we would greatly appreciate it if the reviewer could point out specific instances of ambiguity in the notations. We have already taken note of questions 1-4 and clarify these points in the following response. If there are additional areas of confusion or if any specific notations require further clarification, please bring them to our attention, and we will make the necessary adjustments to enhance the clarity of our work.
>
> ---
>
> **Reference:**
> [1] Li, Daliang, and Junpu Wang. "Fedmd: Heterogenous federated learning via model distillation." arXiv preprint arXiv:1910.03581 (2019).
> [2] Orekondy, Tribhuvanesh, Bernt Schiele, and Mario Fritz. "Knockoff nets: Stealing functionality of black-box models." Proceedings of the IEEE/CVF conference on computer vision and pattern recognition. 2019.
> [3] Kariyappa, Sanjay, and Moinuddin K. Qureshi. "Defending against model stealing attacks with adaptive misinformation." Proceedings of the IEEE/CVF Conference on Computer Vision and Pattern Recognition. 2020.
> [4] Tramèr, Florian, et al. "Stealing machine learning models via prediction {APIs}." 25th USENIX security symposium (USENIX Security 16). 2016.

---

> ### Author Response · Authors · 2023-11-20
> **Response to Question 1-5**
>
> **Question 1:**
> >Why does the accuracy on the private dataset express the success rate of the model stealing attack? What if the attack mode is very generalized which achieves high accuracy in the same data distribution?
>
> **Response:**
> As also mentioned in *Response to Weakness 1-2*, we identify the local model itself as the privacy element that we aim to attack and defend. To assess the success of the attack, we employ a widely adopted metric in model stealing attacks, which is the functionality of the model. Specifically, we evaluate whether the attacking model can achieve high classification accuracy on the defender’s classification task.
>
> We acknowledge the reviewer's point regarding the possibility of a very generalized attack achieving high accuracy in the same data distribution. In response to this concern, we have conducted an ablation study in a non-iid setting, and the results are reported in *Response to Weakness 4 and Question 1* to Reviewer kvRw. Even in the non-iid setting, we observe that AdaMSA achieves similar performance compared to the victim model, indicating the effectiveness of our attack.
>
> This analysis enhances the robustness of our evaluation for the attack performance, and we will incorporate these considerations into our revised manuscript.
>
> **Question 2:**
> >Since $D_{pub}$ is unlabeled, how to quantify Eq. 1 for $D_{mix}$?
>
> **Response:**
> Our focus is not on calculating $D_{mix}$ directly; rather, our objective is to derive the bound of the privacy risk, which is quantified by the performance of the attacking model. The bound we aim to obtain is independent of the ground truth label $f_p$. The unlabeled nature of $D_{pub}$ does not affect our ability to establish a meaningful privacy risk bound.
>
> **Question 3:**
> >The construction of $D_{mix}$ is based on the empirical sets of $D_{priv}$ and $D_{pub}$. How does it reflect the true distribution of $D_{priv}$ and $D_{pub}$?
>
> **Response:**
> Our primary objective is not to precisely replicate the true distribution but rather to systematically manipulate the correlation between the private and mix datasets. Creating such a controlled environment enables us to evaluate the impact of varying correlations on the privacy risks in logit-based Federated Learning (FL). While the constructed $D_{mix}$ may not precisely reflect the true distribution, it serves as a valuable tool for assessing the privacy risk of logit-based FL methods through changes in the correlation between public and private data.
>
> In our revised manuscript, we will emphasize the purpose of our empirical construction of $D_{mix}$ and its role in providing insights into the sensitivity of logit-based FL methods to variations in correlation between private and public datasets. We appreciate the reviewer's consideration of this aspect and thank you for bringing this to our attention.
>
> **Question 4:**
> >Since the adversary cannot touch $D_{priv}$, how to construct $D_{mix}$?
>
> **Response:**
> We construct $D_{mix}$ with the aim of systematically manipulating the correlation between the private and public datasets. It's important to clarify that, in our setting, $D_{mix}$ functions as the public dataset in the standard logit-based FL setting [1]. In other words, all clients and the server possess the mix dataset as the unlabeled public data described in Section 3.1.
>
> **Question 5:**
> >What is the difference between the theoretical analysis of the work compared to Blitzer et al., 2007?
>
> **Response:**
> Our theoretical analysis is primarily based on the work by Blitzer et al., 2007. Our initial objective is to derive the bound of the privacy risk in our specific setting. We establish that this risk can be effectively measured by evaluating the performance of the attacking model. Subsequently, we derive the privacy bound based on the principles and findings from prior works.
>
> ---
>
> **Reference:**
> [1] Li, Daliang, and Junpu Wang. "Fedmd: Heterogenous federated learning via model distillation." arXiv preprint arXiv:1910.03581 (2019).

---

### Official Review · Reviewer_nzQ8 · 2023-10-31

**Soundness:** 3 good
**Presentation:** 2 fair
**Contribution:** 3 good
**Rating:** 6
**Confidence:** 3

**Summary:**

The paper investigates the hidden privacy risks in logit-based Federated Learning (FL) methods through a blend of theoretical and empirical approaches. It introduces the Adaptive Model Stealing Attack, which utilizes historical logits in training and provides a theoretical analysis of the associated privacy risk bounds. They also propose a defense strategy that perturbs the transmitted logits in the direction that minimizes the privacy risk while maximally preserving the training performance. Experiments under different settings demonstrate the performance of the proposed attack and defense.

**Strengths:**

* The paper provides the first analysis of the hidden privacy risk in logit-based FL methods. An attack and a corresponding defense method are proposed to quantify and prevent the privacy risk. The authors also provide a theoretical bound for the privacy risk.
* Experiments under different FL settings have been conducted to demonstrate the performance of the attack and the defense.

**Weaknesses:**

* The computational complexity of the proposed attack and defense are not discussed.
* For the defense method, the approximation error of the proposed heuristic solver is not analyzed.
* The number of communication rounds of the experiments is small. It would be interesting to see whether the performance of the attack and the defense still hold with hundreds or thousands of communication rounds.

**Questions:**

* For the proposed attack, how to determine the threshold $T_0$? Why the importance weight $w$ is linearly dependent with $t$ (rather than exponential dependence for example)?
* How many federated clients are in the experiments?

---

> ### Author Response · Authors · 2023-11-20
> **Response to Weakness 1-3**
>
> **Weakness 1:**
> > The computational complexity of the proposed attack and defense are not discussed.
>
> **Response:**
> We appreciate the reviewer‘s valuable suggestions regarding the computational complexity of the proposed attack and defense algorithms.
>
> Assume the communication round is $M$. The computational complexity of AdaMSA described in Algorithm 1 is $O(M|\mathcal{D}_{pub}|)$. The additional computational complexity of proposed defense is $O(dim(p))$, where $dim(p)$ is the dimension of logit $p$.
>
> We will add the computational complexity analysis in the Appendix B.
>
> **Weakness 2:**
> > For the defense method, the approximation error of the proposed heuristic solver is not analyzed.
>
> **Response:**
> We are grateful for the reviewer‘s insightful comment regarding the absence of an analysis on the approximation error of the proposed heuristic solver for the defense method. In the current phase of our research, our primary emphasis has been on identifying and quantifying the privacy risk in logit-based FL. The experimental results of our proposed solver for the defense method represent the initial steps in addressing the optimization problem outlined in Eq. (6) and (7).
>
> We acknowledge the reviewer's observation and recognize the necessity for an enhanced solution to the optimization problem, including a thorough examination of the approximation error associated with the proposed heuristic solver. We value the importance of evaluating both the efficiency and accuracy of the solver as integral components of our ongoing work.
>
> The reviewer's suggestion aligns with our commitment to refining and extending our research. In our latter revised manuscript, we will explicitly point out the need for an improved solver, considering both efficiency and accuracy, and highlight this as a specific avenue for future research.
>
> **Weakness 3:**
> > The number of communication rounds of the experiments is small. It would be interesting to see whether the performance of the attack and the defense still hold with hundreds or thousands of communication rounds.
>
> **Response:**
>
> We appreciate the reviewer's comment regarding the number of communication rounds in our experiments. The consideration of the impact of communication rounds is indeed crucial for understanding the robustness and scalability of our proposed attack and defense methods.
>
> In our experiments, we observed that the local models converge at approximately round 6-7 in the close-world setting and at round 20 in the open-world setting. Given these observations, we made a conscious decision to select 10 communication rounds for the close-world setting and 30 rounds for the open-world setting. These choices were made to capture the convergence points while maintaining a balance between computational efficiency and experimental completeness.
>
> Furthermore, we attempted to extend the training to 100 rounds in the open-world setting. However, we found that the performance showed only minor variations within a range of 1-2%. This indicates that, beyond a certain point, additional communication rounds did not significantly impact the observed results.

---

> ### Author Response · Authors · 2023-11-20
> **Response to Question 1-2**
>
> **Question 1:**
> > For the proposed attack, how to determine the threshold $T_0$? Why the importance weight is linearly dependent with $t$ (rather than exponential dependence for example)?
>
> **Response:**
> We appreciate the reviewer's questions regarding the determination of the threshold $T_0$ and the linearity of the importance weight $w_t$ in our proposed attack. Your insights prompt valuable clarifications that we aim to address in our revised manuscript.
>
> For $T_0$, in the current paper, we have fixed $T_0$ as 3 to report the main results presented in Table 2. The choice of $T_0$ was made empirically based on observations in our experimental settings. We found that $T_0$ set to 3 yielded optimal results, striking a balance between the server's storage requirements for historical logits and the effectiveness of the attack. In our revised manuscript, we will provide additional context on this choice and discuss the impact of threshold $T_0$ in further detail, as outlined in Table 3.
>
> Regarding the linearity of the importance weight $w_t$, we have empirically observed that the adoption of a simple linear dependence for the importance weight $w_t$ yields effective results in our experiments. We appreciate the suggestion to add discussions on the impact of the hyperparameter setting. In our latter revised manuscript, we will explore potential implications of using different forms of dependence, such as exponential dependence.
>
> **Question 2:**
> > How many federated clients are in the experiments?
>
> **Response:**
> We follow the setting in [1] and set the number of clients as 10 in our experiments.
>
> ---
>
> **Reference:**
> [1] Li, Daliang, and Junpu Wang. "Fedmd: Heterogenous federated learning via model distillation." arXiv preprint arXiv:1910.03581 (2019).

---

### Official Review · Reviewer_kvRw · 2023-11-05

**Soundness:** 3 good
**Presentation:** 3 good
**Contribution:** 2 fair
**Rating:** 5
**Confidence:** 3

**Summary:**

The paper studies the privacy risk in logit-based federated learning (FL). In particular, the authors provide theoretical analysis to bound the privacy risks and propose a model stealing attack adapted to the logit-based FL settings. In addition, the authors also provide a defense strategy that perturbs the transmitted logits to minimize privacy risks.

**Strengths:**

1. The paper is well written.
2. The experiment presentation is clear.
3. The defense is simple and effective.

**Weaknesses:**

1. First of all, I have some questions and doubts about the significance of logit-based FL in the community of FL. I have checked the logit-based FL papers mentioned in the related work, and they are impactful on the FL community. Currently, logit-based FL seems not to be a well-established and standard norm in FL. From this perspective, studying the privacy risks of logit-based FL is unlikely to have an impact on the community in the long run.
2. The tricks used in the proposed attack lack technical depth. The proposed attacks improve by the previous baseline via a temporal weighted factor, making the attack an incremental improvement.
3. Ony baseline (MSA) is too naive. To demonstrate the effectiveness of the tricks used in the proposed attack, I suggest adding more baselines—for example, no threshold $T_0$ or setting $w_t=1$ in the proposed attacks.
4. Non-iid setting of FL. In Figure 1, the author states that the server aims to infer client $k$’s private models. I wonder if the attack makes sense in the non-iid setting of FL or if client $k$ is a poisoned client. In this case, the objective of the attack should be also justified.

**Questions:**

1. Is Adaptive attack possible in the presense of the attack knows the defense (e.g., obfuscate the logit with added noise)?

---

> ### Author Response · Authors · 2023-11-19
> **Response to Weakness 1-3**
>
> **Weakness 1:**
> > First of all, I have some questions and doubts about the significance of logit-based FL in the community of FL. I have checked the logit-based FL papers mentioned in the related work, and they are impactful on the FL community. Currently, logit-based FL seems not to be a well-established and standard norm in FL. From this perspective, studying the privacy risks of logit-based FL is unlikely to have an impact on the community in the long run.
>
> **Response:**
> Thank you for your concern regarding the significance of logit-based FL in the community of FL. we address this concern in the General Response 1.
>
> **Weakness 2:**
> > The tricks used in the proposed attack lack technical depth. The proposed attacks improve by the previous baseline via a temporal weighted factor, making the attack an incremental improvement.
>
> **Response:**
> Thank you for your insightful comments regarding the technical depth of the proposed attack. While we acknowledge that our proposed AdaMSA can be seen as an incremental improvement of MSA, we would like to highlight a key aspect of our contribution. One of our primary objectives is to **identify and quantify the privacy risk in logit-based FL**. This task is nontrivial due to the unique characteristics of logit-based FL, where only logits on public dataset are available, and existing privacy attacks against FL fail to work, as demonstrated in Appendix A.
>
> To address this challenge, we propose a methodology that first steals the client’s private model and then the stolen model is evaluated on the client’s private test dataset to **provide a measurable privacy metric for logit-based FL**. Based on the unique characteristics of logit-based FL, we then propose AdaMSA, aiming to design a more informative attacking logit for the attacking model to learn from. We also provide a detailed comparison of the difference between existing MSA and our proposed attack in Section 2.2.
>
> **Weakness 3:**
> > Only baseline (MSA) is too naive. To demonstrate the effectiveness of the tricks used in the proposed attack, I suggest adding more baselines—for example, no threshold $T_0$ or setting $w_t=1$ in the proposed attacks.
>
> **Response:**
> We appreciate the reviewer's suggestion to include additional baselines to further demonstrate the effectiveness of the tricks used in the proposed attack. We value your feedback and would like to clarify our baselines in the paper.
>
> As the first study to identify and quantify this privacy risk in logit-based FL, we chose to establish a baseline by employing a naive MSA modified with a threshold setting ($T_0=0$; $w_T=1$, otherwise 0). As presented in Table 2, the simplicity of this baseline allows for a clear comparison and understanding of the impact of our proposed attack.
>
> To evaluate the impact of threshold T_0, we have discussed the impact of the threshold setting ($T_0$) in Table 3. We will add discussions on the impact of the hyperparameter setting $w_t$ in our latter revised manuscript.

---

> ### Author Response · Authors · 2023-11-19
> **Response to Weakness 4 and Question 1**
>
> **Weakness 4:**
> > Non-iid setting of FL. In Figure 1, the author states that the server aims to infer client k’s private models. I wonder if the attack makes sense in the non-iid setting of FL or if client k is a poisoned client. In this case, the objective of the attack should be also justified.
>
> **Response:**
> We appreciate the reviewer's insightful comment regarding the non-iid setting of Federated Learning (FL) and the potential situation of a client being a poisoned client.
>
> For **non-iid setting**: We conduct more ablation experiments to investigate the influence of the non-iid setting. We split the training data according to a Dirichlet distribution following [1]. The non-iid level of data is controlled by the Dirichlet parameter $D$. We report the primary result in Openworld-CF setting to illustrate our observation. We observe that: 1) increasing the non-iid level (i.e. decrease $D$) will decrease the utility of local model as well as the attacking model performance; 2) AdaMSA achieves similar performance compared to the victim model, indicating AdaMSA is still effective in the non-iid setting.
>
> | **Defense** | $D=1.0$ | | $D=100$ |   |
> |:-:|:-:|:-:|:-:|:-:|
> | | Victim | AdaMSA | Victim | AdaMSA |
> | **Unprotected** |  $66.32\pm0.79$ | $66.26\pm1.05$ | $71.09\pm1.21$ | $71.21\pm0.99$ |
> | **Cross-domain** | $49.04 \pm 1.83$ | $49.11 \pm 1.64$ | $54.24\pm1.92$ |  $54.20\pm1.65$|
> | **One-shot** | $57.30\pm1.46$ | $57.42\pm1.02$ | $63.06\pm0.55$ | $63.25\pm0.97$ |
> | **DP-G** | $59.48\pm1.55$ | $60.01\pm0.87$ |  $65.88\pm1.43$ | $66.89\pm1.03$ |
> | **DP-L** | $59.38\pm2.12$ | $59.90\pm1.52$ | $65.59\pm1.96$ | $65.99\pm1.97$ |
>
> More ablation experimental results on the influence of the non-iid setting will be included in the Appendix of later revised manuscript.
>
> For **malicious client:** In this paper, our primary assumption is that clients are honest, and the server is semi-honest, as demonstrated in Table 1. We recognize that exploring other scenarios where the client is semi-honest or even malicious could provide valuable insights into the robustness and privacy of logit-based FL methods. Your suggestion raises an important aspect that we acknowledge and consider as a valuable direction for future work.
>
> **Question 1:**
> > Is Adaptive attack possible in the presence of the attack knows the defense (e.g., obfuscate the logit with added noise)?
>
> **Response:**
> We appreciate the reviewer‘s insightful question regarding the adaptability of the attack in the presence of additional knowledge about the defense, particularly the use of obfuscating logits with added noise. This is indeed an interesting consideration that we would like to address.
>
> In our current study, we make the assumption that the adversary only has knowledge of the victim‘s logits on public data. However, we acknowledge that if the adversary possesses additional knowledge, such as awareness of the defense method or even the type of noise added on the logits, it could potentially develop a more powerful attack by leveraging denoising techniques to reverse the input logits and against the obfuscation-based defense [2-3].
>
> ---
> **Reference:**
> [1] Hsu, Tzu-Ming Harry, Hang Qi, and Matthew Brown. "Measuring the effects of non-identical data distribution for federated visual classification." arXiv preprint arXiv:1909.06335 (2019).
> [2] Balle, Borja, and Yu-Xiang Wang. "Improving the gaussian mechanism for differential privacy: Analytical calibration and optimal denoising." International Conference on Machine Learning. PMLR, 2018.
> [3] Nasr, Milad, and Reza Shokri. "Improving deep learning with differential privacy using gradient encoding and denoising." arXiv preprint arXiv:2007.11524 (2020).

---

> > ### Comment · Reviewer_kvRw · 2023-11-23
> > **Response after Rebuttal**
> >
> > Thanks for the authors' response. After reading the rebuttal, I still share the same concerns as Reviewer K1GZ on the motivation of the work. My concern on the technical depth also remains the same after rebuttal. Thus, I will keep my score.

---

### Author Response · Authors · 2023-11-19
**General Response**

We appreciate the reviewers’ thoughtful comments and concerns regarding the significance of logit-based Federated Learning (FL) in the broader FL community. We would like to address these concerns and provide additional context to the broader implications of logit-based FL and our work.

**Response 1: Significance of logit-based Federated Learning**

While we acknowledge that parameter-based FL methods, such as FedAvg, currently dominate the field, novel approaches, including logit-based FL methods, are continuously being explored to overcome the limitations of existing FL techniques.

The primary goal of logit-based FL method [1] was to address the well-known limitations of parameter-averaging FL approaches [2]:
- The constraint of using the same model architecture across all clients.
- High communication costs associated with exchanging model parameters and updates.
- The privacy leakage issues arising from such exchanges.

Following this direction, numerous variants of logit-based FL methods [3-5] were proposed as potential solutions to mitigate these limitations. These methods have further catalyzed a broad application of logit-based FL in industrial scenarios [6-8].

**Response 2: Significance of our work**

The variants of logit-based FL [3-5] can be viewed as the combination of parameter-based and logit-based FL. Similar to parameter-based FL approaches, these methods inevitability suffer from the existing gradient-based attacks [9-10]. The pure logit-based FL methods, that only exchange logits on public data, are generally believed as safe to share against the existing privacy attacks in FL [3,11,12]. In this paper, we focus on this scenario and identify the first potential privacy risk within such scheme.

We appreciate the reviewers’ engagement and feedback, and we are committed to refining our manuscript to ensure its meaningful contribution to the FL research community. We look forward to further discussions and incorporate these considerations into our revised paper.

---
**Reference**
[1] Li, Daliang, and Junpu Wang. "Fedmd: Heterogenous federated learning via model distillation." arXiv preprint arXiv:1910.03581 (2019).
[2] Mora, Alessio, et al. "Knowledge distillation for federated learning: a practical guide." arXiv preprint arXiv:2211.04742 (2022).
[3] Gong, Xuan, et al. "Ensemble attention distillation for privacy-preserving federated learning." Proceedings of the IEEE/CVF International Conference on Computer Vision. 2021.
[4] Lin, Tao, et al. "Ensemble distillation for robust model fusion in federated learning." Advances in Neural Information Processing Systems 33 (2020): 2351-2363.
[5] Zhu, Zhuangdi, Junyuan Hong, and Jiayu Zhou. "Data-free knowledge distillation for heterogeneous federated learning." International conference on machine learning. PMLR, 2021.
[6] Liu, Tian, et al. "Efficient Federated Learning for AIoT Applications Using Knowledge Distillation." IEEE Internet of Things Journal 10.8 (2022): 7229-7243.
[7] Sui, Dianbo, et al. "Feded: Federated learning via ensemble distillation for medical relation extraction." Proceedings of the 2020 conference on empirical methods in natural language processing (EMNLP). 2020.
[8] Li, Beibei, et al. "Incentive and Knowledge Distillation Based Federated Learning for Cross-Silo Applications." IEEE INFOCOM 2022-IEEE Conference on Computer Communications Workshops (INFOCOM WKSHPS). IEEE, 2022.
[9] Zhu, Ligeng, Zhijian Liu, and Song Han. "Deep leakage from gradients." Advances in neural information processing systems 32 (2019).
[10] Zhao, Bo, Konda Reddy Mopuri, and Hakan Bilen. "idlg: Improved deep leakage from gradients." arXiv preprint arXiv:2001.02610 (2020).
[11] Bouacida, Nader, and Prasant Mohapatra. "Vulnerabilities in federated learning." IEEE Access 9 (2021): 63229-63249.
[12] Hu, Hongsheng, et al. "Membership inference attacks on machine learning: A survey." ACM Computing Surveys (CSUR) 54.11s (2022): 1-37.